# Soil slope monitoring with Distributed Acoustic Sensing under wetting and drying cycles

Jiahui Kang<sup>1,2</sup>, Fabian Walter<sup>1</sup>, Tobias Halter<sup>1,3,4</sup>, Patrick Paitz<sup>5</sup>, and Andreas Fichtner<sup>3</sup>

Correspondence: Jiahui Kang (jiahui.kang@wsl.ch)

Abstract. Hydromechanical soil response to moisture variations reflects complex subsurface dynamics that are critical for geoengineering, slope stability, and other soil health-related fields. While laboratory experiments have provided insights into soil behavior under varying wetness and loading conditions, field-scale observations with high spatial and temporal resolution remain limited. In this study, we present a 2 month field monitoring approach using Distributed Acoustic Sensing (DAS), which enables high-resolution, full-coverage, and continuous monitoring of a grass-covered soil slope. DAS allows for subsurface characterization and time-lapse monitoring of soil moisture dynamics using ambient noise interferometry. Furthermore, by analyzing nanostrain-scale deformation in conjunction with stress state derived from in situ soil moisture measurements, we demonstrate that DAS can track real-time volumetric changes in response to both long-term and daily cyclic moisture variations. We suggest DAS as a valuable tool for the continuous detection of moisture-driven changes in soil mechanical properties with high resolution.

#### 1 Introduction

Soil mechanical behavior is governed by complex interactions between environmental factors, including moisture dynamics, temperature fluctuations, and stress conditions. Understanding soil mechanics is critical across different applications. For instance, rainfall can trigger shallow landslides through rapid water infiltration and pore pressure buildup (Iverson, 2000). In geotechnical engineering, understanding soil mechanical responses is crucial for ensuring the long-term stability of infrastructure such as road embankments, dams, and foundations (Gens, 2010). Climate change affects different environmental variables, especially rainfall and temperature, posing new challenges for hazard mitigation (Diffenbaugh and Field, 2013; Gariano and Guzzetti, 2016; IPCC et al., 2023).

To characterize soil behavior under varying environmental conditions, the effective stress principle provides a theoretical framework for assessing soil stability through its stress state (Terzaghi, 1943). Under ideal linear elastic conditions, effective stress and strain exhibit a proportional relation, governed by elastic moduli such as the shear modulus, Young's modulus, and bulk modulus, which reflect soil stiffness. However, natural soils rarely behave as purely elastic materials. At very small

<sup>&</sup>lt;sup>1</sup>Swiss Federal Institute for Forest, Snow and Landscape Research, Zurich, Switzerland

<sup>&</sup>lt;sup>2</sup>Faculty of Geosciences and Environment, University of Lausanne, Lausanne, Switzerland

<sup>&</sup>lt;sup>3</sup>Department of Earth and Planetary Sciences, ETH Zurich, Zurich, Switzerland

<sup>&</sup>lt;sup>4</sup>WSL Institute for Snow and Avalanche Research SLF, Davos Dorf, Switzerland

<sup>&</sup>lt;sup>5</sup>Grün Stadt Zurich, Zurich, Switzerland

strains (nano- to microstrain), soil behaves elastically. As strain increases, nonlinear elasticity dominates (micro- to millistrain), while at higher strains (> millistrain), plastic deformation occurs, leading to irreversible changes such as strength degradation (Hardin and Drnevich, 1972; Fredlund and Rahardjo, 1993). In the field, repeated changes in stress state due to environmental loading can introduce residual plastic strain in the soil. Over time, the accumulation of plastic strain from these cyclic effective stress changes can lead to progressive soil degradation and eventual failure - a phenomenon known as long-term fatigue (Dif and Bluemel, 1991; Li and Selig, 1996).

The effective stress-strain behavior of soil is influenced by environmental factors, with moisture and temperature being two dominant external drivers. Studies have examined their effects experimentally and numerically. Shear-infiltration tests on residual soils demonstrate that slope failure is primarily associated with a loss of matric suction, reducing the soil's shear strength (Melinda et al., 2004). Dong et al. (2020) established a connection between soil shrinkage, effective stress, and compression characteristics under drying conditions employing an incremental linear elasticity approach. Mechanical stresses and strains can also be produced by thermal gradients, which cause differential expansion and contraction within the soil matrix. Such deformations modify the soil's pore structure and consequently alter its water retention characteristics (Rotta Loria and Laloui, 2017).

While laboratory studies have advanced our understanding of soil behavior, real-world conditions present complexities that these controlled environments cannot fully replicate. The disparity in scale between laboratory samples and field conditions can also influence the applicability of findings. Field monitoring of soil status employs a variety of methods, including geotechnical instrumentation (Ding et al., 2000; Carri et al., 2017), remote sensing technologies (Colesanti and Wasowski, 2006; Wang et al., 2023), and more recently, the application of optical fiber-based sensing systems (Simeoni and Mongiovì, 2007; Wu et al., 2019; Zheng et al., 2020). However, few studies focus specifically on effective stress-strain relations and the real-time monitoring of soil properties. These include point measurements showing pyroclastic grain failure under drying and wetting cycles (Pasculli et al., 2017) and thermal fatigue in loess slopes due to daily heating and cooling cycles potentially contributing to progressive slope degradation over time (Lan et al., 2021). Understanding of soil behavior in the field remains limited, as different sites exhibit different influences on soil hardening or degradation.

Seismic methods offer an attractive alternative to in situ point measurements for monitoring soil mechanical properties. They primarily rely on the analysis of shear wave velocity  $(v_s)$ , which is sensitive to variations in soil density, moisture content, and stress state (Stokoe and Santamarina, 2000; Hussien and Karray, 2016). Constructing virtual seismic sources from ambient seismic noise thus enables temporal monitoring of the subsurface  $v_s$  structure (e.g., Boness and Zoback, 2004; Mainsant et al., 2012; Larose et al., 2015) and thus subsurface characterization through surface wave inversion (e.g., Renalier et al., 2010; Mordret et al., 2013). The capabilities of seismic monitoring can be magnified with Distributed Acoustic Sensing (DAS), which probes the seismic wavefield via strain or strain rate along a fiber-optic cable with nanostrain sensitivity and meter-scale resolution (Parker et al., 2014; Zhan, 2019; Lindsey and Martin, 2021). Extensive research has already focused on ambient noise analysis using DAS (e.g., Martin and Biondi, 2017; Dou et al., 2017; Ajo-Franklin et al., 2019; Rodríguez Tribaldos and Ajo-Franklin, 2021; Fichtner et al., 2023). This shows that DAS offers new opportunities to provide first-of-their-kind

observations of small-magnitude soil deformation that are highly resolved in both space and time and distributed over a field-scale extent.

In this study, we deployed a fiber-optic cable for two months on a grass-covered sandy loam slope. We observe soil stiffening, indicated by an increasing temporal gradient between effective stress and strain, as well as cycles of daily variations. The observed soil "breathing" featuring daytime contraction and nighttime expansion indicates that hydrological processes rather than thermal effects dominate soil deformation. Our work thus highlights the use of DAS for soil slope monitoring beyond the laboratory scale.

## 2 Hydromechanical processes in soil slopes

65

#### 2.1 Soil shrinkage and swelling within the effective stress framework

The infiltration of water into soil leads to spatially and temporally variable changes in the stress state of the soil (Bogaard and Greco, 2016). These stress state modifications, particularly in the near-surface zone, can precede and contribute to slope instability by altering the mobilized shear strength and effective normal stress distributions along potential slip surfaces (Bishop, 1959; Iverson, 2000; Bogaard and Greco, 2016). Building on the effective stress framework, a unified stress model that captures the interactions between air, water, and soil particles across both saturated and unsaturated conditions is expressed as (Lu et al., 2010)

$$P_e = \sigma - P_a - \sigma_s,\tag{1}$$

where the effective stress  $(P_e)$  is a function of the overburden stress  $(\sigma)$ , the atmospheric pressure  $(P_a)$ , and the suction stress  $(\sigma_s)$ . The suction stress represents interparticle forces acting on soil grains, such as capillary and van der Waals attraction, which depend on wetness conditions and soil texture (Lu et al., 2010).

Approximations of soil wetness states are commonly obtained by measuring two key parameters: volumetric water content (VWC) using electromagnetic sensors and soil water potential (SWP) using tensiometers. VWC quantifies the ratio of water volume to the soil's total bulk volume. SWP reflects how tightly water is held in the soil matrix by measuring the energy required to move water from the soil to a reference state of free water at atmospheric pressure (Wicki and Hauck, 2022). SWP measurements encompass three components: matric potential, pressure potential, and solute potential, where matric potential reflects the tension exerted by soil particles on water due to capillary forces (Or et al., 2011). The solute potential is negligible for naturally precipitated water. SWP measurements directly represent matric potential in unsaturated soil and pressure potential in saturated soil (Wicki et al., 2023). The combination of VWC and SWP measurements can be used to derive suction stress (Lu et al., 2010):  $\sigma_s = -S_w * (u_a - u_w)$ , where  $u_a$  and  $u_w$  are the pore air pressure and pore water pressure, respectively, and SWP is a direct measurement of their difference  $-(u_a - u_w)$ . In unsaturated soil where  $u_a > u_w$ , SWP is negative, while in saturated condition, SWP approaches zero or becomes positive (Fig. 1). The effective soil water saturation  $S_w$  can be derived as:  $S_w = \frac{\theta - \theta_r}{\theta_s - \theta_r}$ , where  $\theta$  is the measured VWC,  $\theta_r$  the residual water content and  $\theta_s$  the saturated water content (Wicki et al., 2023; Halter et al., 2024).

**Figure 1.** Schematic representation of soil shrinkage and expansion with the simplified equation of effective stress neglecting the atmospheric pressure. Increased effective stress due to evapotranspiration leads to soil shrinkage, while reduced stress from water infiltration causes expansion when suction stress dominates at shallow depths.

Variations in effective stress alter the mechanical forces acting on soil grains, causing shrinkage under increased stress and expansion under reduced stress (Fig. 1). In the near-surface soil layer, drying leads to increasingly negative suction stress, which outweighs the reduction in overburden stress caused by water loss from evapotranspiration. As effective stress increases, soil particles are drawn closer together, reducing pore space and leading to volumetric contraction (shrinkage). Conversely, during rainfall, suction stress increases and can even become positive as the soil saturates. This causes a net reduction in effective stress, and the reduced interparticle forces allow grains to move apart, increasing pore space and causing expansion. The transition between saturated and unsaturated conditions is critical, as rapid changes in suction stress can lead to abrupt reductions in soil strength which can lead to slope failures (Fredlund and Rahardjo, 1993).

## 2.1.1 Cyclic wetting and drying cycles

100

105

In addition to the imminent change of stress state, unsaturated surface soils are subject to wetting-drying cycles where daytime evapotranspiration leads to drying, and nighttime rehydration causes wetting (Gens et al., 2006; Diel et al., 2019; Ng and Zhan, 2007). These cycles induce mechanical fatigue, where repeated expansion and contraction cause progressive plastic deformation, leading to irreversible changes in soil structure. Over time, this degradation manifests as reduced shear strength, increased porosity, and microcrack formation (Dif and Bluemel, 1991; Hall, 1999; NG and ZHOU, 2014; Tang et al., 2016, 2020). This effect can compromise the long-term stability of slopes, even in the absence of major external triggers such as extreme rainfall events (Petley, 2004). However, geotechnical monitoring techniques such as inclinometers, extensometers, and strain gauges, typically limited to microstrain resolution, often fail to detect these small but progressive changes.

Spatial heterogeneity on the slope further complicates effective stress distributions. Differences in soil properties like texture, porosity, permeability, and sun exposure create localized variations in suction stress, altering the way moisture infiltrates and redistributes within the slope (Hicks and Samy, 2002; Le et al., 2015). Capturing these spatially complex interactions requires high-resolution and continuous monitoring techniques that transcend the limitations of traditional point-based methods.

## 110 3 Study Site and Measurements

## 3.1 Napf-Emmental

120

125

Our study site is located on a steep, grassy slope with inclinations of up to 33° in Wasen im Emmental, in central Switzerland (Fig. 2a). This region, situated in the foreland basin north of the Alps, is part of the Napf Formation, consisting of upper freshwater molasse deposits that formed during the Oligocene and Miocene epochs. Geologically, these deposits are composed of conglomerate layers intersected by marlstone, overlain by Quaternary talus deposits and a fractured soil layer (Stähli et al., 2011; Christian et al., 2019). As a result of intense fluvial erosion since the Pleistocene, the topography of the Napf region is characterized by steep slopes. During soil moisture sensor installation, soil samples revealed a vertical textural gradient based on USDA soil taxonomy developed by the United States Department of Agriculture and the National Cooperative Soil Survey: the upper layer (0.13 m depth) is classified as sandy loam, transitioning to loam at greater depths (0.53–0.98 m) with increasing clay content (Wicki et al., 2023; Halter et al., 2024). The site's pre-Alpine setting influences its meteorological conditions, particularly its precipitation patterns. Long-term meteorological data (1960–2020) shows that most of the annual precipitation occurs between May and August, sometimes accompanied by intense rainfall events and thunderstorms (Wicki et al., 2023). These precipitation events, combined with permeable soils, promote rapid water infiltration, leading to an increase in pore water pressure, and potentially to slope failure (Stähli and Wicki, 2021; Bisanti et al., 2005). The Napf region's susceptibility to rainfall-induced landslides is well-documented. Historical records analyzed by Halter et al. (2024) identified 632 landslide events in the area since 2000, the majority of which were triggered by rainfall. These findings highlight the critical need to investigate the interplay between precipitation, soil hydrology, and slope stability in this region to better understand and predict landslide hazards.

### 3.2 DAS and soil moisture measurements

In May 2023, we installed a fiber-optic gel-filled non-metallic loose tube cable that was connected to a Silixa iDAS interrogator in a trailer parked in a nearby farmhouse garage (Fig. 2b). From the garage to the road, the cable is enclosed in a protective tube. The trench installation begins at the gravel road, with the cable routed underground in a ca. 0.1-0.2 m deep trench along the cow pasture with slopes between 29° and 33° (Fig. 2b). The DAS system operates with a channel spacing of 1.02 m and a 400 Hz sampling rate, with cable strain rates averaged over a 10 m gauge length (Parker et al., 2014). The continuous recording period lasted from 22 July to 20 September 2023. However, the cable was damaged beyond ch234, likely due to

**Figure 2.** Overview of the study site in the Napf region, central Switzerland. (a) A digital elevation model (Federal Office of Topography swisstopo, 2024a) of the Napf region shows the study site, marked by the red triangle. The inset map indicates its location within Switzerland. (b) A digital color orthophotomosaic of the study slope (Federal Office of Topography swisstopo, 2024b) highlights the DAS cable installation, shown in blue, running from the DAS interrogator (near the farmhouse) to the top of the slope. Key locations include the loafing shed and the locations of soil moisture sensors EMM\_1 and EMM\_2 (denoted by white and green diamonds, respectively). The red dashed line indicates the broken section of the DAS cable beyond channel 234, where only noise is recorded. (c) The altitude profile along the DAS cable provides elevation changes along the installation route. Channels 36, 180, and 234 are marked to illustrate key locations along the cable in (b) and (c).

grazing cattle or installation-related conditions. The measuring distance is divided into three sections based on topography and surface conditions:

- 1. **Section A** (**ch0-36**): Climbing the gradient of the lowest slope region (ca. 33°), characterized by relatively dense and uniform grass coverage.
- 2. **Section B** (**ch37-180**): Perpendicular to the slope along a narrow, one-foot-wide pathway formed by repetitive farming activities. The path shows varying grass cover and compaction, with shadows from adjacent trees reducing sunlight exposure in the eastern part during the morning hours. At approximately ch45, the cable transitions from a west-southward to an east-southward slope orientation, coinciding with observable changes in grass coverage.

3. **Section C** (**ch181-234**): Perpendicular to section B (ca. 29°). The toe of the section is also shaded by trees. Compared to section A, the grass coverage in section C is sparser and less uniform.

Since April 2019, point-measurements of soil moisture have been conducted at a 10 min interval near the top of the slope, close to the malfunctioning cable section (EMM\_1), and in a flat area adjacent to the loafing shed at the slope toe (EMM\_2) (Fig. 2b) (Wicki et al., 2024). VWC was derived from dielectric permittivity measurements following Topp et al. (1980), using capacitance-based sensors (ECH2O 5TE, METER Group). SWP was recorded with tensiometers (T8 Tensiometer, METER Group), which measure pressure differences in the soil with a piezoelectric sensor embedded in a water-filled porous ceramic cup. At EMM\_1, two sensors of each type (2 × VWC and 2 × SWP) were installed at depths of 0.15 m, 0.30 m, 0.50 m, and 1.00 m. At EMM\_2, two sensors of each type were installed at 0.15 m, 0.50 m, and 0.95 m, with an additional sensor pair (1 × VWC and 1 × SWP) installed at 0.20 m and 0.70 m. No site-specific calibration of the sensors was conducted, as the original study by Wicki et al. (2024) focused primarily on relative changes in VWC. While this study used absolute values to estimate effective stress, only relative changes in effective stress were analyzed for comparison with the strain rates derived from the DAS measurements. We computed daily medians to match the dv/v analysis. For low-frequency strain analysis, we used the 0.15 m depth, while the dv/v models were built using a multi-depth composite at 0.15 m and 0.95 m. Further details are provided in the following sections.

#### 4 Data

#### 160 4.1 DAS signals

The DAS measurements were decomposed into signals above and below 1 Hz to capture both dynamic and quasi-static slope deformation. Signals above 1 Hz were utilized for surface wave analysis to derive soil layer velocity profiles (e.g., Dou et al., 2017; Ajo-Franklin et al., 2019; Rodríguez Tribaldos and Ajo-Franklin, 2021; Shen et al., 2024). Recent studies have shown that DAS, with its high spatial and temporal resolution, can detect hidden landslide processes with strain signals below 1 Hz (Ouellet et al., 2024). These include subtle, long-duration millimeter-scale displacements that conventional methods cannot capture.

The primary sources of signals above 1 Hz include cow and farmer movement, as the fiber-optic cable traverses a grazing area. Figure 3a illustrates the distinct cow signals originating at approximately ch180. These high-amplitude signals result from ground impacts across hundreds of Hz and generate surface waves that travel along section B. The amplitude of these direct impacts attenuates rapidly over short distances (ca. 20 m) but the surface waves remain detectable further along the cable within a frequency range of 6–28 Hz (Fig. 3b). The impact-induced surface waves, coupled with their clear move-out provide valuable data for surface wave inversion and characterization of the soil layer structure.

The strain rate below 1 Hz reveals three signal types (Figs. 3c-d). The first type demonstrates diurnal cyclic variations characterized by positive strain rate during nighttime periods and negative strain rate during daytime periods, with magnitudes typically ranging from -10 to +10 nm/m/s. Additionally, there are short-duration positive strain rate values during the daytime.

Figure 3. Overview of DAS measurements recorded from 23–25 July 2023, highlighting signal characteristics above and below 1 Hz. (a) The raw data show cow ground impacts, which cause surface waves propagating along the cable. (b) The frequency spectrum analysis of the raw data in panel (a) illustrates the distribution of signal energy across all channels. (c) The strain rate dynamics below 1 Hz exhibit cyclic patterns, with positive strain rates occurring during nighttime (17:00–07:00 UTC) and negative strain rates during daytime (07:00–17:00 UTC). There are also positive strain-rate values occurring at multiple channels over short time periods. Cow generated quasi-static signals are enlarged in the circle, demonstrating their long-lasting impact on strain rate measurements. (d) The hourly precipitation data highlights strong DAS signals during rainfall events at 04:00 and 16:00 UTC on 24 July.

These short-duration signals are most prominent during daytime hours, where they are superimposed on the broader negative strain-rate background. The third signal type is recorded during precipitation events (Figs. 3c-d), with pronounced spatial heterogeneity and elevated magnitudes exceeding  $\pm 10$  nm/m/s. The DAS system also detects quasi-static signals generated by cow movements, analogous to vehicle-induced signals described in van den Ende et al. (2023). These cow-induced signals, while not central to the analysis of signals below 1 Hz, represent a notable noise source, resulting in local strain rates of over  $\pm 100$  nm/m/s.

**Figure 4.** Soil moisture measurements at 0.15 m depth at EMM\_1 and EMM\_2 during recording period. Panels show (a) volumetric water content (VWC), (b) soil water potential (SWP), (c) soil temperature, and (d) merged hourly precipitation combining in situ and CombiPrecip product data.

## 4.2 In situ Soil Moisture Measurements

Soil point measurements (VWC, SWP, temperature, and precipitation) were collected at multiple depths at locations EMM\_1 and EMM\_2. We focus on measurements at 0.15 m depth, corresponding best to the fiber-optic cable installation depth of approximately 0.1–0.2 m. The data from the point measurements were recorded at a 10 min interval and are shown in Figs. 4a-c. During the monitoring period, the VWC and temperature sensors at EMM\_1 malfunctioned after 13 August 2023 (Figs. 4a and c), and the rain gauge ceased operation on 12 September 2023. To maintain continuity in precipitation monitoring, hourly precipitation data from the CombiPrecip product, which integrates rain-gauge and radar estimates (Gabella et al., 2017; Germann et al., 2022), were combined with the in situ rain gauge measurements (Fig. 4d). Minor data gaps in soil moisture measurements, typically lasting several tens of minutes and caused by temporary sensor clogging, were addressed using linear interpolation.

Soil moisture closely tracked rainfall: VWC increased during infiltration, while SWP became more negative. Measurements from EMM\_1 and EMM\_2 were highly correlated for both variables (Pearson's r > 0.9), indicating that either site can serve as a representative indicator of regional soil-moisture dynamics.

**Figure 5.** Data processing workflow for surface wave tomography. (a) A representative 2 s DAS record shows direct cow impacts and the resulting surface waves. (b) Next, we calculated cross-correlations between ch75 and ch165 and stacked them over one hour (dashed vertical lines). (c) The following dispersion analysis includes manually edited dispersion curves from the correlation functions. (d) The dispersion curves were inverted for shear-wave velocity structure, and we here present the 200 best-fitting solutions, along with the mean and optimal models. Lower misfit values correspond to better agreement with the observed data.

# 195 5 Seismic velocity model profiling

#### 5.1 Surface wave inversion

To leverage cow-induced seismic signals for surface wave tomography, we focused on 90 DAS channels in section B, as sections A and C provided fewer effective channels, limiting aperture and therefore velocity profile resolution (Park et al., 1999; Yilmaz, 2001). Figure 5 illustrates our workflow for deriving the shallow subsurface velocity structure.

We followed established processing steps for the DAS data (Rodríguez Tribaldos and Ajo-Franklin, 2021). This included detrending and demeaning to eliminate systematic biases, followed by downsampling to 200 Hz to focus on the frequency band of interest while maintaining computational efficiency. A zero-phase Butterworth bandpass filter (1-90 Hz) isolated surface wave components while preserving phase information. Temporal amplitude normalization using a 1 s running absolute mean effectively suppressed transient high-amplitude events. Spectral whitening was applied to balance frequency content and enhance phase coherence across the bandwidth of interest. Cross-correlation analysis was performed with ch165 as the virtual source. To optimize the signal-to-noise ratio, we implemented a two-stage stacking approach: first, we averaged the time-symmetric (causal and acausal) components of 1 min correlation windows, and second, we applied robust stacking over 1 h intervals (Pavlis and Vernon, 2010).

The farmer rotated grazing locations every few days to allow grass recovery, which leads to intermittent grazing sessions near section B. When cows were present, their movement generated strong and long-lasting low-frequency signals, as demonstrated in Fig. 3c. By screening daily low-frequency signal plots, we identified 9 days with grazing events. To minimize potential biases from location-specific cow activity, for each of these days, we manually selected the 1 h window where cow-induced signals were distributed along the cable rather than concentrated in specific locations for dispersion analysis. Noise cross-correlation yielded both fundamental and higher-order Rayleigh modes (Fig. 5c). We selected the fundamental mode and the first two higher modes. This choice was guided by their consistent presence across all dispersion curves from the 9 days of data. To invert for the subsurface velocity structure, we utilized the Geopsy software package, which employs a hybrid neighborhood algorithm for robust parameter estimation (Wathelet et al., 2004).

We tested the model configuration with 2–5 layers. The three-layer model with an additional half-space yielded the lowest misfit, indicated by the percentage error between the modeled and observed dispersion curves. Lower misfit values correspond to better agreement between modeled and observed data. The two-layer model, though yielding a comparable misfit, estimated a first-layer depth of 2.8 m, while the three-layer model estimated 1.53 m, which closely matches depths determined via manual hand auger drilling (1.5–2.0 m depth) and electrical resistivity tomography near EMM\_1 (Wicki and Hauck, 2022; Wicki et al., 2023). Therefore, we adopted a three-layer  $v_s$  model for nine days of monitoring presented in Fig. 7a. While the effects of changing noise sources may affect the retrieval of surface waves using cross-correlations (Hanasoge, 2012), the apparent  $v_s$  profiles remain consistent across different days, suggesting that variability from grazing patterns has a negligible influence. The mean of the top 100 best-fitting models across the nine-day inversion results reveals velocity transitions at 1.53 m, 4.2 m, and 12 m. The first layer (0–1.53 m,  $v_s$  = 108 m/s) corresponds to a soft soil layer characterized by unconsolidated sandy loam and loam. The second layer (1.53–4.2 m,  $v_s$  = 220 m/s) marks a transition to a mechanically stronger material. Given the geological setting of the Napf region, this layer is interpreted as a coarse, highly weathered molasse conglomerate. The third layer (4.2–12 m,  $v_s$  increasing from 220 to 320 m/s) likely represents more consolidated Molasse conglomerate (Labhart, 2009; Anbazhagan and Sitharam, 2009). Full inversion results for all tested models, including sensitivity analyses, are provided in Appendix A.

**Figure 6.** (a) This panel shows the daily stacked cross-correlation on 22 July 2023, with the reference trace from ch80 highlighted in brown. The coda wave window from 0.3 s to 1.2 s ensures separation from direct arrivals. (b) This panel illustrates the temporal evolution of coda waves extracted from ch80 throughout the monitoring period.

# 5.2 Time-lapse monitoring: dv/v

235

240

We monitored shear wave velocity changes using coda wave interferometry (CWI). Coda waves, generated by scattered wave energy, are sensitive to small changes in seismic velocity because of their extended propagation paths and increased interaction with the medium compared to direct waves (Snieder, 2006; Larose and Hall, 2009). For our analysis, 1 min cross-correlation segments were stacked into daily cross-correlation stacks. We focused on ch80 for each day with ch165 as the virtual source because of its clear separation between direct arrivals and coda waves (Fig. 6a).

The stretching technique (Sens-Schönfelder and Wegler, 2006) was applied to estimate relative seismic velocity changes dv/v between consecutive days. Figure 6b illustrates the coda waves in cross-correlation stacks at ch80 between 0.3 s and 1.2 s throughout the monitoring period. We applied a parameter range of  $\pm 5\%$  and implemented a moving window analysis with 0.03 s steps and 0.3 s window lengths. The final relative velocity change dv/v was derived from the median values across all time windows. The frequency range chosen for the analysis was 8-16 Hz, which corresponded to the strongest fundamental mode signals, as shown in Fig. 5.

To interpret the relation between CWI-derived dv/v and soil moisture variations, we applied a rock physics framework, which models shear wave velocity in unconsolidated soils (Solazzi et al., 2021; Shen et al., 2024). This model considers an unconsolidated soil system composed of n homogeneous and isotropic layers with varying compositions. The shear wave

velocity  $(v_{s,j})$  of the  $j^{th}$  layer can be expressed as

$$v_{s,j} = \sqrt{\frac{\mu_j}{\rho_{b,j}}}. (2)$$

Here,  $\mu_j$  is the effective shear modulus of the layer, and  $\rho_{b,j}$  is its effective bulk density. Both parameters depend on soil moisture conditions. Under steady-state saturation conditions,  $\mu_j$  is equal to the drained shear modulus  $\mu_{m,j}$  using Biot-Gassmann poroelastic theory (Biot, 1941; Gassmann and für Geophysik, 1951) and can be derived as follows with the Hertz-Mindlin (HM) contact model (Mindlin, 1949)

$$\mu_{j} = \mu_{m,j} = \frac{2 + 3f - (1 + 3f)\nu_{s,j}}{5(2 - \nu_{s,j})} \left[ \frac{3N^{2}(1 - \phi_{j})^{2}\mu_{s,j}^{2}}{2\pi^{2}(1 - \nu_{s,j})^{2}} P_{e,j} \right]^{\frac{1}{3}},$$
(3)

where  $\mu_{s,j}$  is the effective shear modulus of the solid grain,  $\nu_{s,j}$  is Poisson's ratio of the soil matrix,  $\phi_j$  is the porosity,  $P_{e,j}$  is the effective stress within the layer, and f and N are empirical soil structure constants. The effective bulk density is given by

$$\rho_{b,j} = (1 - \phi)\rho_{s,j} + \phi[S_{w,j}\rho_w + (1 - S_w)\rho_a]. \tag{4}$$

Here,  $\rho_{s,j}$ ,  $\rho_w$ , and  $\rho_a$  are densities of the soil grains, water, and air, respectively, while  $S_{w,j}$  denotes the effective water content which can be derived from VWC measurements (Wicki et al., 2023; Halter et al., 2024). In Eqs. 3 and 4, the parameters  $\rho_{s,j}$ ,  $\mu_{s,j}$ ,  $\nu_{s,j}$ , and  $\phi_j$ , along with constants f and N, are detailed in Appendix B, as they are not affected by short-term soil moisture fluctuations. The key variables governing moisture-dependent variations in shear wave velocity are  $P_{e,j}$  given in Eqs. 1 and  $S_{w,j}$ . During rainfall, the reduction in effective stress leads to a decrease in shear modulus, while the increase in water content raises the effective bulk density. Together, these factors contribute to the decrease in modeled  $v_{s,j}$ .

Next, we considered continuous soil moisture data from EMM\_2 at 0.15 m and 0.95 m depths. Given the 1.53 m inverted soil thickness, we approximated it as a two-layer system, with the upper 0.15 m representing the first layer and the lower 1.38 m the second. The full derivation of the velocity change profile is provided in Appendix B.

The daily dv/v monitoring results from both CWI and the rock physics model (RPM) are presented in Fig. 7b, alongside daily precipitation. The CWI-derived dv/v fluctuations remain within approximately  $\pm 1\%$ , demonstrating sensitivity to soil moisture dynamics. The correlation analysis yields a Pearson coefficient between CWI- and RPM-derived dv/v of 0.51, testifying to the dependence of CWI-derived dv/v fluctuations on soil moisture variations.

## 6 Low-frequency DAS strain

## 6.1 Data processing

To capture variations in both time and space domains, we computed the mean DAS strain rate over 10 channels, corresponding to the gauge length, with a 5-channel overlap. A median filter with a 31 s window was applied to each 10-channel average to mitigate cow-related quasi-static noise. We then integrated the filtered strain rate over time to obtain the accumulated strain changes relative to the initial reference point at 22 July 2023, 00:00:00.

Figure 7. (a) One-dimensional  $v_s$  models for nine monitoring days show consistency, with each curve representing a specific day (color-coded). (b) The temporal comparison of dv/v derived from coda wave interferometry (black solid line) and rock physics model (red dashed line) is shown alongside daily precipitation (blue bars). Key wetting and drying phases are highlighted with blue and orange arrows, respectively. The vertical gray lines denote the boundaries of the defined periods (T1–T5), corresponding to distinct soil moisture conditions to be discussed in the discussion section.

#### 6.1.1 Instrumental drift quantification

The analysis of long-term accumulated strain requires careful consideration of potential instrumental drift. To quantify this, we used the cable section that remained isolated from ground deformation, looped, and hung on a pin within the garage hosting our interrogator. This section serves as a reference to isolate instrument-related effects from ground strain. Figure 8a shows the accumulated strain change for both the buried sections (A, B, and C) and the garage section. A consistent monotonic decrease in strain, superimposed with intraday cyclic variations, is evident across all channels. The strain variations among the 80 channels within the garage (Fig. 8a, green lines) are highly coherent with minimal time shifts.

To isolate long-term strain change from diurnal fluctuations, we applied Seasonal-Trend decomposition via LOESS (STL) (Cleveland et al., 1990) to the averaged strain from both the buried and garage sections. This method separates the time series into trend, daily periodic, and residual components. The resulting long-term trend of the garage section exhibits a linear decrease (Fig. 8b). A linear fit to this trend reveals a constant instrumental drift rate of -7532 nanostrain/day.

The long-term trend of the buried cable section deviates from this linear pattern, suggesting that it records both the instrumental drift and non-linear ground deformation. The subsequent analysis of the buried cable data will be presented after correcting for instrumental drift.

Figure 8. Relative strain variations. (a) displays the relative strain across DAS channels. The colored lines (blue-to-yellow gradient) represent the 10-channel averaged strain for the buried section. The dark green lines show the strain of the 80 channels of the garage section, with their average plotted in pink. The variation among the channels is so minimal that the individual green lines are nearly indistinguishable and overlap. (b) shows the long-term relative strain trends for both cable parts. The dashed cyan line is a linear fit to the average strain of the garage section, with the daily change indicated in the top right. (c) provides a zoomed-in view of the daily variations (August 12-19, 2023) for the garage section. The plot compares the daily residual strain (pink line) with the air temperature (orange line). The right axis quantifies the cable's thermal sensitivity in terms of equivalent strain every 1° C, indicated with pink dashed line. Shaded regions indicate nighttime (18:00–08:00+1 UTC). (d) shows daily variations for the buried section over the same period. The plot compares the daily residual strain (black line) with the ground temperature measured at 0.15 m depth (orange line). The thermal-corrected strain is shown in a dashed black line. For (c) and (d), shaded regions indicate nighttime (18:00–08:00 UTC).

## **6.1.2** Temperature effects on the fiber-optic cable

DAS signals below 1 Hz are influenced by both strain changes along the cable and temperature effects (Bakku, 2015; Gao et al., 2018; Leggett et al., 2022; Sidenko et al., 2022). Temperature effects can introduce bias into strain measurements if not accounted for. As such, analyzing and correcting for thermal effects is critical for reliable interpretations of strain variations. We first assessed this empirically by comparing the daily residual strain of the garage section with direct air temperature measurements (Fig. 8c showing the period between August 12 and 19, 2023). This comparison shows a high correlation. We calculated the ratio of the daily strain to the daily temperature change. This yielded an observed apparent temperature sensitivity of within  $\pm 1 \times 10^{-2}$  millistrain/°C. However, as the garage's thermal environment is different from the open air, this section cannot serve as a source for direct quantitative correction of the buried cable.

To quantify the theoretical impact of temperature on the buried cable, we followed the approach of Leggett et al. (2022). Adopting the parameter settings of Carr et al. (1990), we calculated the equivalent strain variation induced by temperature fluctuations using the relation

$$\Delta \varepsilon = \frac{\frac{\xi_T}{n} + \frac{\xi_{\varepsilon}}{n} \alpha + \alpha}{1 + \frac{\xi_{\varepsilon}}{n}} \Delta T. \tag{5}$$

Here,  $\Delta \varepsilon$  represents the equivalent strain,  $\Delta T$  is the temperature change, and the choices of coefficients  $(\xi_T, \xi_\varepsilon, n, \text{ and } \alpha)$  are shown in Table C1 in Appendix C.

In this study, ground temperature measurements were taken every 10 mins at EMM\_2 at 0.15 m depth. Daily temperature variations were within 1° C, inducing a strain change of about  $1.1 \times 10^{-2}$  millistrain. This value is similar to the sensitivity observed in the garage section and is approximately one order of magnitude smaller than the primary daily strain variations measured in the soil.

We applied a thermal correction to the buried cable data by subtracting the calculated temperature-induced strain. The result is shown in Fig. 8d, which compares the final, temperature-corrected daily residual strain with the uncorrected strain. The two curves are nearly identical, confirming that the influence of direct temperature changes on the fiber is minimal.

## 6.2 Strain response to environmental variables variations

Figures 9a-c display the 10-channel strain averages after instrumental drift correction for sections A, B, and C, while Fig. 9d illustrates the overall average strain across all channels alongside precipitation data. The temporal changes in strain reveal synchronized trends across space. Using the same temporal divisions as in the dv/v analysis, we examine five distinct periods (T1 to T5). A consistent trend of decreasing strain, indicative of soil contraction, is observed during periods T2 and T4. Additionally, intraday cyclic strain variations are characterized by daytime contraction and nighttime expansion (Fig. 8d). During rainfall events (T1, T3 and T5), the general decreasing trend is disrupted, and the strain levels are either sustained or increased, reflecting relative soil expansion.

The intraday strain variations (Fig. 8d) contrast with previous findings that attributed soil "breathing" primarily to thermal stress, describing daytime expansion and nighttime contraction in loess slopes (Lan et al., 2021; Collins and Stock, 2016).

Figure 9. Temporal evolution of strain and environmental variables. (a), (b) and (c) display the 10-channel averaged strain across DAS channels for sections A, B, and C, respectively. Strain values are color-coded by channel indices (blue-to-yellow gradient). Bifurcation events are highlighted in sections A and C. (d) illustrates the overall average strain (black line), precipitation (blue bars), and temperature (orange line). (e) shows the effective stress, VWC and SWP. Key drying and wetting phases (T1–T5), consistent with dv/v monitoring, are separated by vertical gray lines.

During periods of increasing temperature, the DAS strain decrease shows that soil contracts. On the other hand, during periods of decreasing temperatures, the increasing cable strain testifies to soil expansion. This inverse relation is evident not only in the diurnal cycles but also in the long-term drying periods T2 and T4, which are accompanied by a temperature increase and soil contraction (Fig. 9d). We interpret these diurnal strain variations as soil "breathing" driven primarily by hydromechanical cycles from daytime evapotranspiration and nighttime rehydration. These cycles modulate capillary and overburden stresses during warm summer days, influencing the soil's stress state (Tang et al., 2016; Rahardjo et al., 2019).

To bring the strain patterns into a hydromechanical relation, we analyzed soil moisture data (VWC and SWP) within the framework of an effective stress model at 0.15 m depth, following the formulation proposed by Lu et al. (2010) (Eq. 1). We focus on 0.15 m depth because it matches the cable burial depth and thus best represents quasi-static strain, which is different from the dv/v analysis. This is because dv/v is sensitive to seismic velocity changes integrated over several meters depth, whereas the low-frequency strain reflects direct near-surface deformation at the fiber depth. We adapted it using VWC and SWP values to calculate each term as follows (neglecting air pressure):

335 
$$P_e = \rho_e gh - \frac{VWC - \theta_r}{\theta_s - \theta_r} SWP$$
 (6)

where  $h=0.15\,m$  is the measurement depth,  $\rho_e$  is the effective density of the soil, calculated as  $\rho_e=(1-\phi)\rho_s+\phi(\frac{VWC-\theta_r}{\theta_s-\theta_r}\rho_w+(1-\frac{VWC-\theta_r}{\theta_s-\theta_r})\rho_a)$  (Eq. 4). Here, the densities  $(\rho_s,\rho_w,\,\rho_a)$  and porosity  $(\phi)$  are given in Table B2 and B1.  $\theta_r=0.559$  is the residual water content from field measurements (Wicki et al., 2023). The saturated water content  $\theta_s$  is taken as the average of max(VWC) and  $\phi$ .

The resulting effective stress time series exhibits both long-term trends (Fig. 9e) and diurnal fluctuations (Fig. 9f). Pearson correlation analysis indicates that strain is highly correlated with effective stress during drying periods T2 and T4, with negative correlations of -0.85 and -0.88, respectively (Table 1). However, during wetting phases, the correlation coefficients vary more widely from -0.06 to 0.69. The same pattern is also observed in the  $\varepsilon$ -T relation. The lower  $\varepsilon$ - $P_e$  correlation during wetting periods likely results from nonlinear strain-stress behavior under transient water flow conditions. During rapid water infiltration, excess pore water pressure is generated, disrupting the steady-state proportionality between stress and strain (Fredlund and Rahardjo, 1993). Despite this, temperature and effective stress are inherently linked by environmental processes. Rainfall events decrease both effective stress and temperature, while temperature influences evapotranspiration rates. This explains why  $\varepsilon$ -T correlation during T2 and T4 is also high, but negative. However, this does not imply that temperature itself is driving strain changes inversely. Rather, both are responding to hydromechanical soil processes.

The inverse relation between strain and effective stress aligns with fundamental soil mechanics principles. During drying, increased suction stress enhances interparticle tension, raising effective stress (Lu et al., 2010). This compression of the soil skeleton leads to volumetric contraction, which we observe as increased contraction strain (Dong et al., 2020). In contrast, during wetting, water infiltration reduces suction stress, decreasing effective stress (Fig. 1). This reduction weakens interparticle forces, allowing soil grains to move apart, leading to expansion and swelling of the soil matrix. Apart from suction stress, drying and wetting affect the overburden stress by altering soil water volume, which in turn influences the effective stress. Interestingly, the measurements show that beyond a certain threshold, the decrease in overburden stress outweighs the increase in suction

**Table 1.** Pearson correlation coefficients between DAS strain  $(\varepsilon)$ , temperature (T) and effective stress  $(P_e)$  measured at EMM\_2 for five distinct periods (T1 to T5), as well as the entire period.

| Variable 1    | Variable 2 | Correlation coefficient |       |       |       |       |           |
|---------------|------------|-------------------------|-------|-------|-------|-------|-----------|
|               |            | T1                      | T2    | T3    | T4    | T5    | $T_{all}$ |
| $\varepsilon$ | $P_e$      | 0.51                    | -0.85 | -0.06 | -0.88 | 0.69  | -0.26     |
| $\varepsilon$ | T          | 0.61                    | -0.89 | -0.15 | -0.93 | -0.25 | -0.21     |
| T             | $P_e$      | 0.55                    | 0.84  | 0.92  | 0.93  | 0.36  | 0.75      |

stress, as observed during the drying period T2 after 23 August (Fig. 9e). During such sustained drying periods, soil contraction ceases, and soil suction represented by the measured SWP stagnates. Returning to the signal types identified in Fig. 3c, we can attribute them to hydro-mechanical processes driven by soil moisture changes at different timescales. The slow, diurnal strain cycles are consistent with effective stress due to day and night moisture change, while the abrupt, high-amplitude signals are direct responses of pore water pressure to infiltration from rainfall. The short-period positive strain-rate values correspond to small, rapid daytime perturbations observed in the SWP data.

Two bifurcation periods are observed in Figs. 9a and c within sections A and C during wetting periods, where strain divergence suggests localized differences in soil saturation, infiltration pathways, or even small-scale soil creep events (Ouellet et al., 2024). A detailed examination of these abrupt strain drops reveals their coincidence with high-amplitude signals attributed to cow activity, as shown in Figs. C1 and C2 in Appendix C. Accordingly, the strain drops are most likely caused by local soil deformation beneath cow hooves.

#### 7 Discussion

#### 7.1 Daily variations in dv/v

While both measured and modeled dv/v values capture the general trend of velocity reduction during wetting periods and recovery during drying (Fig. 7b), the RPM-derived dv/v responds more quickly to rainfall events, whereas the CWI-derived dv/v exhibits a delayed reaction of one to two days. The difference in response times and magnitudes between the two models is primarily due to their different depth sensitivities. As shown by the sensitivity kernel (Fig. A2b), the CWI-derived dv/v in the 8–16 Hz frequency band is influenced by velocity changes throughout the upper 12 m, including both the soil layer and the underlying molasse conglomerate. The RPM is limited to a simplified two-layer soil model extending to 1.53 m, where the moisture changes are more significant. This explains why the RPM predicts larger dv/v fluctuations than CWI. While a more complex, deeper model would be ideal, we do not have the necessary data from large enough depths during the monitoring period.

Specific wetting and drying phases are highlighted in Fig. 7b, and reveal distinct temporal patterns:

- 1. Wetting period 1 (24 July to 7 August 2023, T1): Two significant rainfall events on 14 July and 4 August induced  $v_s$  reductions of approximately 0.5% and 1% relative to the average value, respectively. These velocity reductions indicate a loss of soil stiffness and shear strength due to increased moisture infiltration and pore water pressure (Iverson, 2000; Mainsant et al., 2012; Bontemps et al., 2020). Frequent rainfall throughout this period suppressed recovery, maintaining dv/v below -0.5%.
- 2. Drying period 2 (8 to 25 August 2023, T2): Minimal rainfall allowed soil moisture to decrease and seismic velocities to recover steadily. The nearly 2% rise in dv/v suggests a progressive stiffening of the soil matrix due to drying-induced consolidation and increased effective stress (Cha et al., 2014; Dong and Lu, 2016). Both CWI- and RPM-derived dv/v curves exhibit a stagnation in recovery around 23 August, despite the absence of precipitation until 25 August. We attribute this phenomenon to the effective stress reaching a threshold and starting to decrease (Fig. 9e), at which point the influence of decreasing overburden stress exceeded that of increasing suction stress.
  - 3. Wetting period 2 (26 August to 29 August 2023, T3): The most intense rainfall event of the monitoring period occurred on 28 August, with daily precipitation exceeding 30 mm, causing a sharp 1.5% drop in dv/v, reaching -0.7% relative to the average reference velocity. This rapid velocity reduction suggests a substantial weakening of soil shear strength.
  - 4. **Drying period 2 (30 August to 11 September 2023, T4)**: Following heavy rainfall, dv/v exhibited a steady recovery as the soil underwent drying and mechanical stiffening. A stagnation in stiffening like during T2 is less apparent.
  - 5. Two subsequent rainfall events (12 September 2023 and onwards, T5): Two smaller rainfall events induced dv/v reductions, which were less pronounced than earlier events.

#### 7.2 Effective stress-strain response

395

The interpretation of the stress-strain relation in unsaturated soils to effective stress and soil strength, assumes steady-state flow conditions, where pore pressure and suction remain stable (Fredlund and Rahardjo, 1993). Therefore, we chose the two drying periods T2 and T4. For T2, only data up to 22 August were analyzed to focus on monotonic effective stress variations. We used the effective stress calculated at 0.15 m (Eq. 6) and the average strain over all DAS channels for comparison (Fig. 9d-e). Figure 10a illustrates the temporal evolution of the effective stress-strain relation with two key processes: a general drying trend characterized by increasing effective stress and accumulated shrinkage and cyclic wetting-drying variations occurring on a daily scale. Strain variations are used as a classification criterion, with orange and blue dots indicating contracting and expanding phases, respectively.

We applied STL decomposition to the effective stress and average strain time series to distinguish between the dominant hydromechanical processes influencing the soil. Although the absolute strain magnitudes are underestimated due to the low strain transfer efficiency of the loose-tube cable (Forbriger et al., 2025), this underestimation acts as a consistent scaling factor and does not affect the interpretation of relative patterns. Figures 10b-e present both the long-term effective stress-strain relation and intraday cyclic dynamics for T2 and T4. The trend components exhibit high correlation coefficients of approximately 0.9

for both periods. This underscores the dominant influence of effective stress in driving soil contraction during drying. The gray lines in Fig. 10a represent the long-term effective stress-strain trend, while the gray dashed lines show linear fits, with their respective equations noted alongside.

We observe that as drying progresses, the effective stress within the soil decreases over time, while the contraction strain accelerates (Figs. 10b–c). This nonlinearity also manifests itself in the diurnal variations where the temporal gradient of the effective stress-strain trajectory increases, particularly in the later stages of drying (Fig. 10a). Under the assumption of a homogeneous and isotropic soil moisture distribution our one-dimensional along-cable strains can be considered proportional to volumetric changes. Consequently, the temporal gradient of the effective stress-strain relation can be taken as a proxy for bulk modulus (Dong et al., 2020). The increasing temporal gradient reflects a rise in bulk modulus, indicating soil stiffening due to consolidation (Dong et al., 2020). Soil stiffening results in a greater amount of stress required to induce the same strain. Additionally, the higher temporal gradient of the fitted curve in T2 compared to T4 suggests a reduction in soil stiffness in T4, likely due to the intense rainfall event on 28 August. This rainfall may have caused pore water pressure buildup, weakening the soil's bulk strength (Iverson, 2000). The corresponding sharp drop in dv/v in T3 (Fig. 7) further supports this interpretation, indicating a reduction in shear strength following the rainfall event.

Despite the strong correlation observed between the long-term trends of contraction strain and effective stress (Figs. 10b-c), the intraday cyclic variations exhibit lower correlation coefficients (approximately 0.75; Figs. 10d-e). This discrepancy is also evident in the intraday loops (Fig. 10a), where the effective stress-strain paths during wetting and drying do not fully overlap. One possible explanation for the lower correlation is the spatial variability in the timing of moisture-induced strain response along the slope. Although we initially assumed homogeneous and isotropic conditions for the long-term trend analysis, small-scale variability in soil properties, infiltration capacities, or preferential flow paths could affect intraday soil strain responses (Wicki et al., 2023). To investigate this, we cross-correlated intraday strain variations in each 10-channel segment with the average strain over the entire cable and calculated the relative time shifts corresponding to the cross-correlation zero lag during periods T2 and T4 (Fig. 10f). The resulting spatial pattern shows consistent time shifts across space in the two periods (correlation coefficient = 0.93). The time differences along the channels reflect the heterogeneous and possibly anisotropic nature of the subsurface. Hydromechanical interactions are influenced by such heterogeneities that affect the local response to weather fluctuations. A detailed analysis of these spatial effects requires co-located strain and moisture measurements, which is beyond the scope of this work.

## 8 Conclusions

We observed two hydromechanical processes: (1) progressive soil consolidation during drying periods, resulting in increased bulk stiffness and soil strengthening; and (2) daily cyclic deformation patterns ("soil breathing"), driven by moisture fluctuations between daytime drying and nighttime moisture recovery, which is in contradiction to thermal stress-induced deformation patterns. The "soil breathing" effects are pronounced in the shallow subsurface, where the stress state is easily modulated by various hydrological processes. In the long term, this could affect soil health through soil fatigue. Those observations pro-

**Figure 10.** Effective stress-strain analysis. (a) shows the time evolution of the effective stress-strain relation for T2 and T4. Daily cycles are color-coded, with orange dots representing expansion (wetting-induced unloading) and blue dots representing contraction (drying-induced loading). The STL-decomposed trend is fitted with gray lines, while dashed lines indicate best-fit curves with their respective equations displayed. (b) and (c) present the long-term trend relations between effective stress and contraction strain for the two drying periods, while (d) and (e) depict the cyclic variations over time. The Pearson correlation coefficients for each relation are provided in the top-left corner of the respective subplot. (f) shows the spatial distribution of relative time shifts between intraday strain variations at each 10-channel segment and the overall averaged strain. Positive shifts indicate earlier strain responses relative to the average curve.

vide new insights into how laboratory-observed phenomena manifest in natural settings, where boundary conditions are less constrained and environmental forcing is more complex. The long-term soil consolidation is further supported by the dv/v analysis, which independently reflects changes in shear wave velocity linked to moisture variations.

DAS-based soil slope monitoring strategies could benefit from further optimization. Implementing refined signal processing techniques can mitigate localized disturbances in strain, such as cow-induced signals. Moreover, integrating DAS more closely with co-located complementary direct moisture measurement systems could reduce uncertainties and further advance our understanding of temporal lags across space. Exploring alternative DAS cable layouts, including parallel or vertical borehole installations relative to slope gradient, could facilitate multi-dimensional characterization of strain tensors and provide comprehensive insights into subsurface hydrodynamic processes. The 10 m gauge length, a fixed parameter of the iDAS interrogator we used, functions as a spatial moving average over a 10 m segment of soil. It filters out localized, small-scale heterogeneities and improves the signal quality for observing the bulk soil response, but inherently limits the spatial resolution of the strain measurement. This averaging effect is a crucial factor when integrating DAS with traditional point-based instruments. Future near-surface studies targeting more localized phenomena would benefit from deployments using interrogators with a configurable and shorter gauge length.

In conclusion, we integrate traditional seismic wave analysis with continuous monitoring of quasi-static deformation using DAS. This enables direct field-scale observations of soil mechanical response, achieving meter-scale along-cable resolution with a 10 m gauge length.. The ability of DAS to capture strain changes across multiple temporal scales offers new opportunities for validating and refining slope behavior models that incorporate both immediate and long-term deformation mechanisms.

Code and data availability. The raw DAS dataset comprises several terabytes and is too large to share in full. We provide the hourly cross-correlation results, a downsampled 1 Hz version of the dataset, soil moisture measurements, and all scripts necessary to reproduce the figures in this paper at Zenodo: https://doi.org/10.5281/zenodo.15191409. The analysis was performed in Python (v3.11.4), using Numpy (Harris et al., 2020), scipy (Virtanen et al., 2020), statsmodels (Skipper and Josef, 2010), and jupyter (Kluyver et al., 2016).

Author contributions. JK conceptualized the study of soil layer analysis via surface wave analysis and low-frequency signals, conducted the data analysis, and drafted the manuscript. FW supervised the project, supported the DAS data acquisition, contributed to the manuscript structure, and provided critical revisions throughout the writing process. TH provided the soil moisture data and contributed to the effective stress–strain analysis and manuscript revision. PP supported the DAS data acquisition and contributed to discussions on data interpretation. AF provided guidance on the surface wave analysis and advised on the interpretation of low-frequency DAS signals, as well as manuscript revision.

Competing interests. The authors declare that they have no conflict of interest.

Acknowledgements. We are thankful to Christoph Wetter, Enrico Bernardini for their help in fieldwork. Spacial thanks to Lorenz Winkel475 mann for generously providing us access to the slope. We also thank Peter Lehmann, Manfred Stähli and James Kirchner for the insightful
discussion. We would like to thank the anonymous reviewer and Sara Sayyadi for their thorough and constructive comments. We are grateful
to the editor, Wolfgang Schwanghart, for his efficient and responsive handling of our manuscript. Our project has received funding from the
European Union's Horizon 2021 research and innovation program (EnvSeis) under grant agreement No. 101073148.

#### Appendix A: Surface wave inversion analysis

Four different layer configurations were tested in the surface wave inversion process, as shown in Fig. A1. The four-layer and five-layer models were excluded due to their high misfit values and inconsistencies across different iterations, which resulted in unrealistic depth variations in the averaged models. Among the remaining two-layer and three-layer models, both achieved similarly low misfit values. The two-layer model exhibited higher consistency across the best 500 inversion iterations, maintaining stable velocity structures at all depths. However, the three-layer model yielded the lowest overall misfit and provided a soil depth estimate that aligned well with field exploration results (Wicki and Hauck, 2022; Wicki et al., 2023). This alignment suggests that a three-layer configuration is the most appropriate for accurately characterizing the subsurface structure.

To evaluate the reliability of the inversion process, the sensitivity analysis was conducted using the Python-based disba library (Keurfon, 2021), which incorporates tools from the Computer Programs in Seismology suite. As shown in Fig. A2, the fundamental mode provides constraints for all three layers' shear-wave velocity and depth with sensitivity around 0.1-0.2. The inclusion of higher Rayleigh modes (Fig. A2), especially mode 2, which provides sensitivity up to 7, was critical for refining the depth of the first soil layer and enhancing the resolution of the shallow structure. The results demonstrate that the combination of the fundamental and higher Rayleigh modes is essential for accurately resolving near-surface features.

**Figure A1.** Surface wave inversion results for different layer configurations. (a)-(d) show the best 500 inversion models for all layer configurations. The cyan line represents the best-fitting model, while the dark cyan line denotes the average model computed from the best 500 iterations. The colorbar indicates the misfit percentage where darker shades correspond to lower misfit values, representing better-fitting models.

Figure A2. Sensitivity analysis for the three-layer velocity model used in surface wave inversion. (a) The input shear wave velocity ( $v_s$ , black) and compressional wave velocity ( $v_p$ , red) profile for the three-layer input model. (b)-(d) Sensitivity kernels for different Rayleigh wave modes (mode 0, mode 1, and mode 2). The color scale represents the frequency range, where blue to yellow shading indicates increasing frequency. The pink-shaded regions correspond to negative velocity perturbations induced by  $v_s$  variations.

#### Appendix B: Description of rock physics model

For a given soil layer j, the Poisson's ratio  $(\nu_{s,j})$  of the soil skeleton is expressed as:

$$\nu_{s,j} = \frac{3K_{s,j} - 2\mu_{s,j}}{2(3K_{s,j} + \mu_{s,j})}$$
 (B1)

where  $K_{s,j}$  and  $\mu_{s,j}$  are the effective bulk and shear moduli of the soil grains. These moduli are further calculated from the individual constituent properties using Hill's averaging formula (Hill, 1952):

$$K_{s,j} = \frac{1}{2} \left[ \sum_{i=1}^{m} \gamma_i K_{s,i} + \left( \sum_{i=1}^{m} \frac{\gamma_i}{K_{s,i}} \right)^{-1} \right]$$
(B2)

$$\mu_{s,j} = \frac{1}{2} \left[ \sum_{i=1}^{m} \gamma_i \mu_{s,i} + \left( \sum_{i=1}^{m} \frac{\gamma_i}{\mu_{s,i}} \right)^{-1} \right]$$
 (B3)

In these equations,  $\gamma_i$  represents the volume fraction of the *i*-th mineral constituent, and  $K_{s,i}$  and  $\mu_{s,i}$  are the bulk and shear moduli of the individual minerals. Finally, the grain density  $(\rho_{s,j})$  is a weighted average of the densities of the mineral constituents:  $\rho_{s,j} = \sum_{i=1}^{m} \gamma_i \rho_{s,i}$ .

**Table B1.** Soil properties of the 2-layer soil profile EMM 2 (Wicki et al., 2023)

| -     | Textural fraction [%] |               | Properties    |            |              |               |   |     |
|-------|-----------------------|---------------|---------------|------------|--------------|---------------|---|-----|
| Depth | Clay                  | Silt          | Sand (50 -    | USDA       | Bulk         | Porosity      | N | f   |
| (m)   | $(

**Figure C1.** Strain variation and 1 Hz DAS data recorded during the wetting period from 29 to 31 July 2023. (a) the strain trends for section A, where a clear divergence into two distinct groups occurred on 29 July with arrows showing the bifurcation. The channel indices are color-coded. (b) the 1 Hz DAS data, capturing strain rate dynamics across all channels. The color bar reflects compression and extension phases, respectively, while the arrow indicates short-duration, high-amplitude cow signals coinciding with specific strain drops, unrelated to geotechnical movements.

**Figure C2.** Strain variation and 1 Hz DAS data recorded during the wetting period from 28 to 30 August 2023, indicating the bifurcation at section C as above.

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
