# Peer review of "Soil slope monitoring with Distributed Acoustic Sensing under wetting and drying cycles"

_EGUsphere, 2025_

## Author Comment (AC1)

Response to RC2
**Soil slope monitoring with Distributed Acoustic Sensing under wetting and drying cycles**
Discussion: https://doi.org/10.5194/egusphere-2025-1725
Comments from the reviewers are given in black.
Our responses are given in blue. *The revisions to be made in the manuscript are given in italic style.*

**General Comments:**

This manuscript presents results from a two-month field deployment using Distributed Acoustic Sensing (DAS) to monitor a soil slope under natural wetting and drying cycles. The study captures both long-term and daily hydromechanical deformation, combining surface wave inversion, coda wave interferometry, and effective stress modeling. Integration with in-situ moisture data reveals soil "breathing" and progressive stiffening during drying. The paper is well written, the methods are clearly described, and the results are well illustrated. The findings are relevant for understanding moisture-driven soil behavior and slope stability. I only have a few concerns regarding the depth sensitivity of the data and the application of the rock physics model to interpret dv/v changes, which I believe should be addressed more clearly before drawing detailed interpretations.

Thank you so much for your constructive comments. Your review has been instrumental in helping us deepen our analysis of the DAS measurements and address important aspects we had previously overlooked. We believe that your suggestions have significantly improved the clarity, completeness, and overall quality of the manuscript.

**Specific comments:**

• **Depth sensitivity of surface wave inversion and soil moisture sensor:**
Please provide more detail about the soil moisture sensors used in the study—specifically, the measurement depths, the type of sensors, the quantities they measure directly, and whether any scaling or calibration is needed to derive VWC. Since the sensors are not co-located with the fiber-optic cable and are installed in different slope settings, what is the justification for focusing on the 0.15 m depth in the comparison? Line 170 states that 0.15 m depth is chosen because of the cable installation depth, but given that DAS measures strain from propagating waves (which integrate energy over depth and wavelength), how is

the physical cable depth directly related to the depth sensitivity of the seismic measurements?

Thank you. We have updated the paragraph beginning at L146 as follows. The soil moisture measurement at 0.15 m depth was selected for calculating effective stress for coupled analysis with low-frequency DAS data. This choice is based on the burial depth of the fiber-optic cable (0.1-0.2 m). The low-frequency DAS signals analyzed here are induced by quasi-static soil deformation in response to environmental loading (e.g., moisture changes) rather than seismic waves.

"""

*Since April 2019, point-measurements of soil moisture have been conducted at a 10 min interval near the top of the slope, close to the malfunctioning cable section (EMM_1), and in a flat area adjacent to the loafing shed at the slope toe (EMM_2) (Fig. 2b) (Wicki et al., 2024). VWC was derived from dielectric permittivity measurements following Topp et al. (1980), using capacitance-based sensors (ECH2O 5TE, METER Group). SWP was recorded with tensiometers (T8 Tensiometer, METER Group), which measure pressure differences in the soil with a piezoelectric sensor embedded in a water-filled porous ceramic cup. At EMM_1, two sensors of each type (2 × VWC and 2 × SWP) were installed at depths of 0.15 m, 0.30 m, 0.50 m, and 1.00 m. At EMM_2, two sensors of each type were installed at 0.15 m, 0.50 m, and 1.00 m, with an additional sensor pair (1 × VWC and 1 × SWP) installed at 0.20 m and 0.70 m. No site-specific calibration of the sensors was conducted, as the original study by Wicki et al., (2024) focused primarily on relative changes in VWC. While this study used absolute values to estimate effective stress, only relative changes in effective stress were analyzed for comparison with the strain rates derived from the DAS measurements.*
"""

In Figure 5d, the surface wave inversion results appear to have limited resolution in the upper ~2 meters, yet dv/v is compared to moisture changes at 0.15 m depth. Can you clarify this mismatch in depth sensitivity? Also, please provide an estimate of the seismic wavelength of the surface waves used for dv/v analysis. For instance, if the dominant frequency is ~10 Hz and shear wave velocity is ~200 m/s, the wavelength would be ~20 m—much deeper than 0.15 m. Do you have sensor data at greater depths to better match the depth sensitivity of the seismic measurements?

In Figure 5d, the inversion results show that the soil layer depth is approximately 1.53 m. Based on that, the CWI-derived dv/v is compared to a two-layer soil model with a total depth of 1.53 m (L255). As noted in our response to the previous comment, the soil moisture measurement at 0.15 m depth is used

exclusively for comparison with the low-frequency DAS data, not with the dv/v results.

However, we realize that it is important to distinguish between the depth sensitivity discrepancy between CWI- and RPM-derived dv/v:

- The CWI-derived dv/v is shown in the sensitivity kernel (Fig. A2). The seismic waves are sensitive to changes in the entire near surface down to ~12 m, integrating the response of both the soil and the bedrock.
- The RPM-derived dv/v is simplified to a two-layer shallow soil model (extending to 1.53 m) because it's driven by our available soil moisture data. We lack data on moisture variations within the deeper weathered bedrock needed to extend this model further.

This difference in depth sensitivity directly explains why the RPM predicts much larger velocity changes than are actually measured by CWI. We have added the following discussion after L325 to clarify this important point.

*"""*

*The difference in response times and magnitudes between the two models is primarily due to their different depth sensitivities. As shown by the sensitivity kernel (Fig. A2) the CWI-derived dv/v in the 8-16 Hz frequency band is influenced by velocity changes throughout the upper 12 m, including both the soil layer and the underlying molasse conglomerate. The RPM is limited to a simplified two-layer soil model extending to 1.53 m where the moisture changes are more significant. This explains why the RPM predicts larger dv/v fluctuations than CWI. While a more complex, deeper model would be ideal, we do not have the necessary data from large enough depths during the monitoring period.*
*"""*

- **Modeling dv/v under the rock physics framework:**

The manuscript outlines how effective elastic properties such as density and shear wave velocity are computed from effective stress, but it remains unclear how the effective stress is derived from the soil moisture profile. Could the authors clarify the exact steps used to convert volumetric water content and soil water potential into effective stress, especially given the complexity introduced by unsaturated versus saturated conditions?

Thank you for the question. We appreciate the opportunity to clarify them. The detailed theoretical background for this calculation is described in Section 2.2, beginning around line 72. The unified equation takes care of both saturated and unsaturated conditions.

To clarify our final calculation, we have added the adapted equations for effective stress after L295:

"""

*We adapted it using VWC and SWP values to calculate each term as follows (neglecting air pressure):*

$$P_e = \rho_e gh - \frac{VWC - \theta_r}{\theta_s - \theta_r} SWP$$

*where h=0.15 m is the measurement depth, $\rho_e$ is the effective density of the soil, calculated as $\rho_e = (1 - \phi)\rho_s + \phi(\frac{VWC - \theta_r}{\theta_s - \theta_r}\rho_w + (1 - \frac{VWC - \theta_r}{\theta_s - \theta_r})\rho_a)$ (Eq. 4). Here, the densities ($\rho_s, \rho_w, \rho_a$) and porosity ($\phi$) are given in Table B1 and B2. $\theta_r = 0.559$ is the residual water content from field measurements (Wicki et al., 2023). The saturated water content $\theta_s$ is taken as the average of max(VWC) and $\phi$.*

"""

Additionally, what reference shear wave velocity model is used in the dv/v modeling? Is it the same velocity model derived from surface wave inversion in Section 5.1? If not, please explain the differences and justification.

In the RPM, the shear wave velocity is derived directly from Eq. 2 based on soil properties (e.g., effective stress). To calculate dv/v, we take the average of output shear wave velocity as reference. For CWI, it analyzes time shifts in the coda waves to directly calculate dv/v. We take the average waveform as reference to calculate the arrival time difference. Therefore, the resulting dv/v for both models are intrinsically aligned with respect to a mean state.

In Line 250, the authors state that the reduction in effective stress dominates during rainfall events. How is this conclusion supported within the model, especially considering that Figure 1 distinguishes suction stress behavior between unsaturated and saturated conditions? How are these different regimes handled in the dv/v modeling? A clearer explanation of how suction stress is represented and transitions across saturation states would help clarify the model's assumptions and limitations.

In this study, we used a unified equation to take care of both saturated and unsaturated conditions as shown in Eq. 1. Our modeling approach handles this transition continuously through the evolution of SWP. We do not need to switch between different equations for the two regimes. The explanation of how suction stress transits is presented at L85 in combination with Fig 1.

- **Figure 7a:** It is confusing to present temporal variation using full shear wave

velocity models, as shown in Figure 7a, given that the observed changes are on the order of ~1%. This is well within the expected uncertainty of the inversion, which appears to be significantly larger. It does not seem reasonable to interpret such small variations as physically meaningful changes in the velocity structure based on these inversion results.

We believe the variations shown in Figure 7a are physically meaningful for the following reasons. First, although the *accuracy* of the inverted velocity model may be low, the *precision* of CWI is exceptionally high since this technique relies on coda wave interferometry rather than the inverted velocity model. Studies have demonstrated that sub-1% velocity changes can be reliably detected and interpreted (e.g., Brenguier et al., 2008; Shen et al., 2024). In our case, the strong coda wave arrivals give us confidence in the reliability of the observed velocity variations. Moreover, the temporal evolution of dv/v exhibits coherent patterns including distinct drops and recoveries in shear wave velocity that coincide with known soil moisture change.

- **Figure 7b:** What kind of smoothing or filtering was applied to the dv/v time series shown in Figure 7b? Could the apparent delay in dv/v response relative to precipitation events be an artifact of the smoothing process rather than a physical lag in the subsurface response?

Thank you for this thoughtful comment. We applied a bandpass filter (8-16 Hz). To demonstrate that the observed time lag is not an artifact of this bandpass filtering, we calculated the dv/v using the unfiltered daily stacked waveforms. As the plot below shows, the dv/v time series derived from both the filtered and unfiltered data show consistent behavior. This confirms that the lag is not an artifact of the data processing.

[Figure]

**Additional References:**

Topp, G., J., D., and A.P., A.: Electromagnetic determination of soil water content: measurements in coaxial transmission lines, Water resources research, 16, 574–582, https://doi.org/10.1029/WR016i003p00574, 1980.

F. Brenguier et al. ,Postseismic Relaxation Along the San Andreas Fault at Parkfield from Continuous Seismological Observations.Science321,1478-1481(2008).DOI:10.1126/science.1160943

Shen, Z., Yang, Y., Fu, X. et al. Fiber-optic seismic sensing of vadose zone soil moisture dynamics. Nat Commun 15, 6432 (2024). https://doi.org/10.1038/s41467-024-50690-6

---

## Author Comment (AC2)

Response to RC1

**Soil slope monitoring with Distributed Acoustic Sensing under wetting and drying cycles**

Discussion: https://doi.org/10.5194/egusphere-2025-1725

Comments from the reviewers are given in black.

Our responses are given in blue. *The revisions to be made in the manuscript are given in italic style.*

**General Comments:**

The manuscript presents a multi-month DAS deployment on a grass-covered slope in central Switzerland. By pairing high-frequency (>1 Hz) ambient noise interferometry with low-frequency (<1 Hz) quasi-static strain measurements, they aim to demonstrate that DAS can be used to "track real-time volumetric changes in response to both long-term and daily cyclic moisture variations". The topic is very timely and relevant to both the DAS and geohazard communities. The integration of surface wave inversion, dv/v monitoring, and low-frequency strain analysis are technically sound. The dataset is extensive and novel, considering the longer-term duration of the low-frequency DAS measurements combined with near-surface moisture sensors. This work is an important contribution and represents a comprehensive overview of the complementary techniques that can be implemented using DAS to inform slope stability monitoring.

However, there are some critical issues that need to be addressed, relating to the author's interpretation of (1) progressive soil consolidation during drying periods, and (2) daily cyclic deformation patterns driven by moisture fluctuations, as follows:

Thank you so much for your constructive comments. Your review has been instrumental in helping us deepen our analysis of the DAS measurements and address important aspects we had previously overlooked. We believe that your suggestions have significantly improved the clarity, completeness, and overall quality of the manuscript.

**Temperature effects:** The authors indicate that the cyclical deformation patterns observed in the low-frequency DAS strain are driven by moisture fluctuations between daytime drying and nighttime moisture recovery, not by temperature variations. The effect of temperature variations are neglected after

estimating that the daily temperature variations (within 1°C) would induce a strain change of about 1.1 x $10^{-2}$ millistrain which is more than two orders of magnitude smaller than the daily strain variations measured by the DAS system. However, this represents an approximation based on the properties of silica-based fiber, and does not account for the response of the DAS interrogator and fibre optic cable (see next point). Further to the above, the cyclical pattern of the low-frequency strain observations occurring across all channels (Figures 3, C1 and C2) as well as the known sensitivity of low-frequency DAS to temperature, suggest that temperature is a likely dominant contributor.

Thank you for raising this important question regarding temperature effects. Our initial temperature correction was applied only to account for the equivalent strain induced in the cable itself. We also found an error in estimating the relative contribution of temperature to cable strain. The 1.1 x $10^{-2}$ millistain ended up only one order of magnitude lower than the daily variation of strain which is around 0.1 millistain. We revised this section accordingly and applied the thermal correction to the average strain signal. Please see Fig. 8 referenced in the response to the next comment.

Regarding the cyclical pattern observed in the low-frequency strain data, we address this in detail in our response to the following comment.

**Interrogator Instrument Response:** The application of low-frequency DAS for monitoring soil slope processes is still emerging. Here, the authors rely on a two-month period of continuous data acquisition using a Silixa iDAS for the measurements, which provides a measurement of strain-rate. However, the reliability and performance of DAS to measure strain and strain-rate over longer periods is still poorly understood. Ouellet et al. (2024) inferred relative displacements from the LF-DAS using another type of DAS interrogator and were able to obtain reasonable comparison with insitu displacement sensors (ShapeArrays) over a ~three-day period. In this study, there are no collocated sensors that support calibration or confirmation of the strain measurements (e.g., strain gauges, inclinometers, survey prisms), which would be important both for the interpretation and justification for the neglect of temperature. The native strain-rate measurements are integrated to derive strain over the duration of the acquisition. However, this also enables the accumulation of potential noise in the strain-rate data to accumulate over time and appear as

drift. The monotonic decrease that is observed in the strain data may be a result of instrument drift, and not representative of true strain. At a minimum, the authors should address this point by including a discussion of the potential of instrument drift or consider relying on the native strain-rate measurements for their analysis and interpretation. It may also be worthwhile to compare the strain-rate measurements with the gradient of the temperature (temporal derivative) over a shorter time interval, for a more careful assessment of the relationship between the two measurements. The monotonic decrease of strain across all channels over the two-month period does not seem credible, considering both the spatial variability of the cable over the slope as well as the temporal variability considering the numerous rainfall events occurring over the period. For example, considering the nanostrain-rate sensitivity of the DAS measurements, gravity-driven processes of the slope over the two-month period with a shallowly buried cable should incur some observations of visible tension and compression in the strain data, aligning with the topographic profile along the length of the cable over the two-month period.

**Response to Comment on Interrogator Instrument Response:**

Thank you so much for pointing this out! We really appreciate the opportunity to improve our manuscript further with instrument response correction. We have revised **Section 6.1** accordingly to address these concerns and added Fig. 8 to support this discussion.  Furthermore, we replaced all the figures afterwards with instrumental drift correction. The change in trend remains clearly visible, which strengthens our analysis of the long-term trend.

*"""*

**6.1.1 Instrumental drift quantification**

[revised manuscript text omitted]

*strain is shown in a dashed gray line. For (c) and (d), shaded regions indicate nighttime (18:00-08:00 UTC).*
"""

**Response on the Monotonic Strain Trend**

We acknowledge that gravity-driven processes would typically produce spatially variable tension and compression. However, the monotonic strain decrease is observed not only on sloped sections (A and C) but also on the flat section B, where the strain change is not aligned with the direction of gravity. We therefore attribute this monotonic strain variation to volumetric deformation of the soil.

**Response on the cyclical pattern related to temperature change**

We agree that thermal-induced stress from the surrounding soil is an important mechanism to consider. However, if soil expansion and contraction were the dominant factor, strain would be expected to correlate positively with temperature as shown in Fig.8c. Our data reveal the opposite (Fig. 8d) and this indicates that thermal-induced soil stress is not the primary driver of these observed variations.

**Cable Instrument Response:** Please include the specifications of the fiber optic cable used in this study. Particularly at low frequencies, the type of cable also plays an important role in the instrument response (e.g. tight-buffered versus loose-tube, see Ouellet et al. 2024). The impact of the cable type on the response should be included in the discussion.

The cable used is gel-filled non-metallic loose tube cable. We have added the discussion of the cable impact at ~L360 as follows:

"""

*Although the absolute strain magnitudes are underestimated due to the low strain transfer efficiency of the loose-tube cable (Forbriger et al., 2025), this underestimation acts as a consistent scaling factor and does not affect the interpretation of relative patterns.*
"""

**Gauge length effects:** A channel spacing of 1 m and gauge length of 10 m is used in this study. Why were these data acquisition settings used? A gauge

length of 10 m could mask any localized changes in moisture. The author's conclusions (L405) that "This enables direct field-scale observations of soil mechanical response at sub-meter resolution" are technically incorrect, considering the settings (1-m channel spacing, 10-m gauge length) used in this study. The impact of the 10-m gauge length on the results should be included in the discussion, notably in comparing or integrating these measurements with point-based sensors, as for the effective stress-strain response.

*The gauge length of iDAS interrogator we used is a fixed parameter. We have changed the phrasing (L405) to "meter resolution" and added this part in the second paragraph of the conclusions:*

*""""*
*The 10 m gauge length, a fixed parameter of the iDAS interrogator, functions as a spatial moving average over a 10 m segment of soil. It filters out localized, small-scale heterogeneities and improves the signal quality for observing the bulk soil response but inherently limits the spatial resolution of the strain measurement. This averaging effect is a crucial factor when integrating DAS with traditional point-based instruments. Future near-surface studies targeting more localized phenomena would benefit from deployments using interrogators with a configurable and shorter gauge length.*
*""""*

**Coda wave interferometry:** The dv/v estimates are computed with daily cross-correlation waveforms. As such, they cannot resolve sub-diurnal moisture cycles and therefore the claim that the author's key observations of "daily cyclic deformation patterns driven by moisture fluctuations" is supported by the dv/v analysis, appears invalid. Further to this, the dv/v are computed in the 8 to 16 Hz frequency range. The fundamental mode sensitivity kernel (Figure A2b.) appears to indicate varying dv/v sensitivity from 0 to 12 m, extending well below the partially saturated zone in the upper metres. The insitu sensors providing moisture measurements only extend up to ~1 m. The rock physics-based model of dv/v relies on a two-layer soil profile extending to a depth of only 1.38 m. Considering the known sensitivity of dv/v to greater depths (from the sensitivity kernel) it seems important to address this discrepancy more thoroughly in a discussion, or improve the model by extending to a similar depth as the dv/v.

Thank you for these insightful comments. We agree that our daily-stacked CWI dv/v analysis cannot resolve the sub-diurnal cycles and revised L396 accordingly: *"The long-term soil consolidation is further supported ... "*

The two-layer soil profile corresponds to the inverted soil depth along the slope. It would require rock moisture variations to extend the RPM model deeper into the weathered rock layer. However, we do not have the necessary data from large enough depths to force such a model during the monitoring period. We believe this discrepancy is critical to explain why the CWI-derived dv/v changes are smaller than the RPM-derived dv/v. We have added the following discussion L325 to clarify this important point.

*"""*

*The difference in response times and magnitudes between the two models is primarily due to their different depth sensitivities. As shown by the sensitivity kernel (Fig. A2) the CWI-derived dv/v in the 8-16 Hz frequency band is influenced by velocity changes throughout the upper 12 m, including both the soil layer and the underlying molasse conglomerate. The RPM is limited to a simplified two-layer soil model extending to 1.53 m where the moisture changes are expected more significant compared to the 12 m depth. This explains why the RPM predicts larger dv/v fluctuations than CWI. While a more complex, deeper model would be ideal, we do not have the necessary data from large enough depths during the monitoring period.*
*"""*

**Specific Comments:**

**Topographic Profile:** It would be helpful to include a topographic profile or elevation cross-section along the cable route. This would improve interpretation of both seismic and low-frequency strain data, particularly in understanding how slope angle and local relief may influence stress distribution, hydrological changes, and strain patterns.

The topographic profile is given in Fig. 2c. We also provide a figure with spatiotemporal plot of strain-rate together with the topographic file in the next response.

**Spatiotemporal Strain-Rate Images:** To support interpretation of the low-frequency DAS data, it would be helpful to include a spatiotemporal

plot of strain-rate over the entire acquisition period. This would help readers visually assess both temporal variability and any spatially coherent patterns.

Thank you for this suggestion. We have prepared the spatiotemporal plot of the strain rate for your review. The plot shows that the dominant temporal variations are coherent across most channels. This supports our argument that the observed changes are driven by a dominant hydromechanical response rather than localized, topography-driven mass movements. The spatial uniformity of the temporal signal is demonstrated in Fig. 8. To maintain the manuscript's focus, we believe that this new figure does not bring added value and prefer to leave it out.

[Figure]

**Figure 3c:** In addition to the cyclic signals (occurring over all channels), rain-induced high amplitude signals (associated with rainfall events) and

quasi-static cow signals, there appears to be a fourth type of signal (positive strain-rate signals occurring at multiple channels over short time periods that are not associated with rainfall events). Please comment on these signals and whether they are attributed to moisture changes or other processes.

We have marked the fourth type of signal on the figure. Those short-period positive strains correspond to small perturbations in SWP measurements and thus also result from moisture change. We added the description around L165 as follows:

"""

*Additionally, there are short-duration positive strain rate values during the daytime. These short-duration signals are most prominent during daytime hours, where they are superimposed on the broader negative strain-rate background.*

[Figure]

"""

We also added the interpretation of those signals after L315:

"""

*Returning to the signal types identified in Fig. 3c, we can now attribute them to hydro-mechanical processes driven by soil moisture changes at different timescales. The slow, diurnal strain cycles are consistent with effective stress variations due to day and night moisture change, while the abrupt, high-amplitude signals are direct responses of pore water pressure to infiltration from rainfall. The short-period positive strain-rate values correspond to small, rapid daytime perturbations observed in the SWP data.*
*"""*

**Section 2.1.1, Line 106** – Spatial Heterogeneity: The authors state that "spatial heterogeneity on the slope further complicates effective stress distributions." However, the key observations of cyclical strain observations appear generally spatially homogenous across channels. Given DAS's advantage in spatial resolution, it would strengthen the paper to highlight any observed heterogeneity in strain or inferred stress response. Do spatial variations in the LF-DAS signal align with known heterogeneity in vegetation, soil type, or moisture content?

Our discussion of heterogeneity focuses on the comparison across two dry periods at around L380 with Fig. 9f. We observe consistent time shifts of intraday strain variations between the two drying periods across multiple channels. This suggests spatial variability which would be valuable to compare with detailed spatial maps of soil type or moisture content. Such maps, however, were not available.

**Section 7.2. Effective stress-strain response**. This section would benefit from greater clarity on the input data and analytical steps. Is the effective stress calculated using Equation (1) based on measured VWC and SWP? Is the associated strain derived from the LF-DAS data, and if so, is this averaged over the full array or selected channel segments? Explicitly stating this would help improve the clarity of this section.

We have modified L352 to improve clarity with:
*"""*
*We used the effective stress calculated at 0.15 m and the average strain over all DAS channels for comparison (Fig. 8d-e).*
*"""*

*To clarify our final calculation on effective stress, we have also added the adapted equations for effective stress after L295:*

*"""*

*We adapted it using VWC and SWP values to calculate each term as follows (neglecting air pressure):*

$$P_e = \rho_e g h - \frac{VWC - \theta_r}{\theta_s - \theta_r} SWP$$

*where h=0.15 m is the measurement depth, $\rho_e$ is the effective density of the soil, calculated as $\rho_e = (1 - \phi)\rho_a + \phi(\frac{VWC - \theta_r}{\theta_s - \theta_r}\rho_w + (1 - \frac{VWC - \theta_r}{\theta_s - \theta_r})\rho_a)$ (Eq. 4). Here, the densities ($\rho_s, \rho_w, \rho_a$) and porosity ($\phi$) are given in Table B1 and B2. $\theta_r = 0.559$ is the residual water content from field measurements (Wicki et al., 2023). The saturated water content $\theta_s$ is taken as the average of max(VWC) and $\phi$.*
*"""*

**Depth sensitivity of dv/v:** Since the dv/v analysis is performed in the 8–14 Hz frequency band, it would be useful to include an estimation of the corresponding sensitivity depth range. This would support statements such as that on Line ~325: "This suggests that seismic waves integrate the water infiltration process throughout the soil profile rather than merely reflecting near-surface saturation." This could be completed by referring to the fundamental mode sensitivity kernel shown in Figure A2. b).

*We have changed this part of discussion as mentioned above at L325.*

**Groundwater Level Information**: Is there any information available on groundwater levels at the site? Even approximate values or nearby hydrological data would help constrain interpretations of the dv/v changes and assess whether infiltration events reach the saturated zone.

*Thank you for your suggestion. A groundwater penetration test was carried out at site EMM_1 several years ago, but the water table exceeded the instrument's measurement limit of approximately 5 meters. To our knowledge, the nearest available piezometer data, from Hasle-Schächli located 12.8 km away, indicate a groundwater level of around 562 m a.s.l.—more than 200 meters below our study site. Based on this*

information, groundwater influence was not included in the present analysis.

**L270:** Ouellet et al. (2024) follow the approach described by Leggett et al. (2022) in their study. It may be more appropriate to cite the original reference of Leggett et al. (2022) here.

Thanks, we have modified this.

**Section 6.2, L285:** "The intraday strain variations (Fig. 8f) contrast with previous findings ..." Figure 8f is difficult to see within the overall figure. Consider making this into a larger figure, complete with axis labels and values for clarity.

We have removed Fig. 8f and added a new figure (Fig. 8d) above.

**L405:** The statement, "In conclusion, DAS integrates traditional seismic wave analysis with continuous monitoring of quasi-static deformation" should be revised to, "In conclusion, we integrate traditional seismic wave analysis with continuous. monitoring of quasi-static deformation using DAS".

We have made this change to L405.

**Technical Corrections:**

**Section 5.2, Line ~225:** "We focused on ch80 for each day because of its clear separation between direct arrivals and coda waves (Fig. 6a)." Which channel(s) was cross-correlated with ch80 to obtain the cross-correlations? It does not appear to be specified in the text.

Ch165 was used as the virtual shot.  We have changed L227 to:

""""

*We focused on ch80 for each day with ch165 as the virtual source ...*
""""

**Appendix C, Table C1.** Index of fraction - should be Index of refraction

Thank you. We have changed it to index of refraction.

As an additional consideration for the authors', it may help to improve the clarity and impact of the manuscript by separating the seismic (>1 Hz) and low-frequency (<1 Hz) analysis into two separate studies. For instance, the extending the dv/v model over a greater depth and focusing on both the near-surface (0 to 2 m) and deeper (2 to 12 m) sensitivity of the dv/v to changes in effective stress represents an important contribution to the field of environmental seismology. Similarly, improving the analysis and interpretation of the low-frequency DAS observations, with a more rigorous evaluation of the temperature effects, alongside the cable and instrument response, represents a novel study. Separating the two analyses could help improve the clarity and impact of the overall findings.

We appreciate this thoughtful suggestion regarding the manuscript's structure. In this study, we chose to present the seismic and low-frequency analysis together because they offer complementary insights. The dv/v analysis provides depth-integrated sensitivity to subsurface velocity changes, while the low-frequency DAS data offer more localized, directionally sensitive strain-rate observations. Together, they enable a more holistic interpretation of moisture-driven processes. Given the current scope and available data, we do not believe the two analyses are substantial enough for separate studies, but we appreciate the reviewer's perspective and will consider this direction for future, more targeted studies.

**Additional References:**

Thomas Forbriger, Nasim Karamzadeh, Jérôme Azzola, Emmanuel Gaucher, Rudolf Widmer-Schnidrig, Andreas Rietbrock; Calibration of the Strain Amplitude Recorded with DAS Using a Strainmeter Array. Seismological Research Letters 2025; 96 (4): 2356–2367. doi: https://doi.org/10.1785/0220240308

---

## Referee Report (RR1)

**Reviewer Report**

Date: September 10, 2025

Reviewer: Sara Sayyadi

**Overall Assessment**

The revision addresses several structural and figure requests. However, three technical areas remain insufficiently justified or documented: (i) the choice and implications of a 10 m gauge length (GL) for low-frequency DAS; (ii) the documentation and alignment of in-situ soil-moisture observations with DAS metrics; and (iii) limits of daily dv/v stacks and depth sensitivity. Addressing these items would materially improve reproducibility and interpretation.

**Major Comments**

**1) Gauge-length choice and effective spatial resolution**

The methods state a 10 m gauge length (GL) and channel-averaging over ~10 m prior to integrating strain-rate. Please justify this choice and quantify how it affects both sensitivity to meter-scale variability and comparisons with point sensors.

**Requested actions:**

- Provide a rationale for selecting 10 m GL (SNR/stability vs spatial resolution).
- Quantify the effective along-fiber spatial response (e.g., GL convolution kernel and its width/FWHM).
- Include a short re-processing test with a smaller GL (e.g., 2.5–5 m) or a forward model, to illustrate amplitude/phase biases for localized signals.
- Revise any language claiming sub-meter resolution; with 1 m spacing and 10 m GL, the smallest resolvable feature is on the order of 10 m along fiber.
- Clarify implications for comparing DAS (spatial average along GL) against point soil-moisture sensors.

**2) Soil-moisture observations and integration with DAS**

The interpretation relies on moisture-driven mechanisms, but the sensor documentation and the coupling to DAS can be made crisper.

**Requested actions:**

• Explicitly list sensor types/models, measured variables (VWC, SWP), depths, sampling cadence, and distance to the cable.

- Describe calibration/QA and any screening of out-of-range values.
- State how soil-moisture series are aligned with DAS metrics: daily medians for dv/v vs sub-daily series for low-frequency strain/strain-rate; interpolation and gap handling.
- Justify the depths used in comparisons (e.g., shallow composite 0.15–0.40 m vs multi-depth 0.15–1.0 m).
- Summarize simple correlations/lag analyses between moisture indices (and effective stress) and DAS metrics, noting spatial variability.
- If one sensor cluster failed at some point, note reliance on the remaining cluster and any implications.

**3) dv/v temporal resolution and depth sensitivity**

dv/v is computed from daily noise stacks; such products cannot substantiate intraday variability. Also, depth sensitivity of the 8–16 Hz band should be explicitly related to the moisture sensors used.

**Requested actions:**

- State explicitly that dv/v is daily and avoid intraday claims unless sub-daily processing is added.
- Provide a concise depth-sensitivity summary for the analyzed band and reconcile with the chosen moisture depths.

**4) Instrument/cable low-frequency response and temperature/drift**

Thermal strain is estimated to be small and no correction is applied. Please also describe potential interrogator/cable LF response and drift controls when integrating strain-rate.

**Requested actions:**

- Specify cable construction (tight-buffer vs loose-tube, jacket, burial details) and discuss implications for LF response.
- Include a brief check comparing strain-rate with the temporal derivative of temperature over selected windows to assess instrument-related contributions.
- Summarize detrending/high-pass choices used to mitigate integration drift and how robustness was verified.

**Minor / Editorial**

- Appendix: correct "Index of fraction" → "Index of refraction."
- After Figure 3 caption: ensure the next paragraph begins with a capitalized "Hourly ...".
- Consider adding cable specs (make/type) in Methods §3.2 for completeness.

• Ensure all instances of resolution claims are consistent with the stated GL and spacing.

**Minor language/clarity – around L180 (soil-moisture paragraph after Fig. 3)**

The current wording is understandable but could be tighter and more precise (e.g., SWP "decreases" → "becomes more negative"), and the historical citation after "Pearson" isn't needed here.

**Suggested replacement text:**

"Soil moisture closely tracked rainfall: VWC increased during infiltration, while SWP became more negative. Measurements from EMM\_1 and EMM\_2 were highly correlated for both variables (Pearson's r>0.9r>0.9r>0.9), indicating that either site can serve as a representative indicator of regional soil-moisture dynamics."

**Consider rephrasing the following sections for clarity and consistency:**

- A) Results Soil-moisture paragraph (after Fig. 3, ~L175–L186). Suggested replacement:
  - Soil moisture closely tracked rainfall: volumetric water content (VWC) rose during infiltration, while soil-water potential (SWP) became more negative. Measurements from EMM $_1$  and EMM $_2$  were highly correlated for both variables (Pearson's r > 0.9), indicating that either site can serve as a representative indicator of regional soil-moisture dynamics.
- B) Methods Soil moisture (Section 4.2 "In situ Soil Moisture Measurements", ~L170–L176 starter lines; drop-in fits immediately after). Suggested replacement:
  - Volumetric water content (VWC) and soil-water potential (SWP) were measured at 0.15–1.0 m depth (10-min cadence) near the buried cable. We screened outliers, computed daily medians to match dv/v stacks, and used both a shallow composite (0.15–0.40 m) and a multi-depth composite (0.15–1.0 m) to assess sensitivity to depth selection.
- C) Conclusion resolution claim (the "In conclusion..." sentence, ~L403–L407; the number 405 appears at line end).

Suggested replacement:

We integrate traditional seismic wave analysis with continuous monitoring of quasi-static deformation using DAS to track moisture-driven hydromechanical changes, achieving meter-scale along-cable resolution with a 10 m gauge length.

---

## Author Response (AR3)

Response to RC2

**Soil slope monitoring with Distributed Acoustic Sensing under wetting and drying cycles**

Discussion: <a href="https://doi.org/10.5194/egusphere-2025-1725">https://doi.org/10.5194/egusphere-2025-1725</a>

Comments from the reviewers are given in black.

Our responses are given in blue. The revisions to be made in the manuscript are given in italic style.

**Overall Assessment**

The revision addresses several structural and figure requests. However, three technical areas remain insufficiently justified or documented: (i) the choice and implications of a 10 m gauge length (GL) for low-frequency DAS; (ii) the documentation and alignment of in-situ soil-moisture observations with DAS metrics; and (iii) limits of daily dv/v stacks and depth sensitivity. Addressing these items would materially improve reproducibility and interpretation.

We thank you for your careful second review and for your constructive comments. We have carefully re-read our revised manuscript in light of your report.

We believe that most of the concerns raised have already been addressed in our revised manuscript and also want to note that some technical comments (e.g., with references to line numbers) appear to correspond to the original submission rather than the revised version. In the responses below, we indicate where the requested clarifications have been incorporated into the current version, and we have bolded the new additions compared to the first revision. We are happy to make additional edits to further highlight them.

**Major Comments**

**1) Gauge-length choice and effective spatial resolution**

The methods state a 10 m gauge length (GL) and channel-averaging over ~10 m prior to integrating strain-rate. Please justify this choice and quantify how it affects both sensitivity to meter-scale variability and comparisons with point sensors.

Requested actions:

• Provide a rationale for selecting 10 m GL (SNR/stability vs spatial resolution).

- Quantify the effective along-fiber spatial response (e.g., GL convolution kernel and its width/FWHM).
- Include a short re-processing test with a smaller GL (e.g., 2.5–5 m) or a forward model, to illustrate amplitude/phase biases for localized signals.
- Revise any language claiming sub-meter resolution; with 1 m spacing and 10 m GL, the smallest resolvable feature is on the order of 10 m along fiber.
- Clarify implications for comparing DAS (spatial average along GL) against point soil-moisture sensors.

The gauge length of the iDAS interrogator is an acquisition parameter (not a post-processing setting). In our deployment it was set to the default of 10 m, which represents a common trade-off between improved signal-to-noise stability and reduced sensitivity to sub-meter heterogeneity. We have already removed all "sub-meter" resolution claims in the first-round revision. The role of gauge length is discussed explicitly as well in L451:

m

The 10 m gauge length, a fixed parameter of the iDAS interrogator we used, functions as a spatial moving average over a 10 m segment of soil. It filters out localized, small-scale heterogeneities and improves the signal quality for observing the bulk soil response but inherently limits the spatial resolution of the strain measurement. This averaging effect is a crucial factor when integrating DAS with traditional point-based instruments. Future near-surface studies targeting more localized phenomena would benefit from deployments using interrogators with a configurable and shorter gauge length.

mm

**2) Soil-moisture observations and integration with DAS**

The interpretation relies on moisture-driven mechanisms, but the sensor documentation and the coupling to DAS can be made crisper.

Requested actions:

- Explicitly list sensor types/models, measured variables (VWC, SWP), depths, sampling cadence, and distance to the cable.
- Describe calibration/QA and any screening of out-of-range values.
- State how soil-moisture series are aligned with DAS metrics: daily medians for dv/v vs sub-daily series for low-frequency strain/strain-rate; interpolation and gap handling.

- Justify the depths used in comparisons (e.g., shallow composite 0.15–0.40 m vs multi-depth 0.15–1.0 m).
- Summarize simple correlations/lag analyses between moisture indices (and effective stress) and DAS metrics, noting spatial variability.
- If one sensor cluster failed at some point, note reliance on the remaining cluster and any implications.

Thanks for your suggestion. Sensor models, depths and calibration details are already in Section 3.2 L146:

um

Since April 2019, point-measurements of soil moisture have been conducted at a 10 min interval near the top of the slope, close to the malfunctioning cable section (EMM\_1), and in a flat area adjacent to the loafing shed at the slope toe (EMM\_2) (Fig. 2b) (Wicki et al., 2024). VWC was derived from dielectric permittivity measurements following Topo et al. (1980), using capacitance-based sensors (ECH2O 5TE, METER Group). SWP was recorded with tensiometers (T8 Tensiometer, METER Group), which measure pressure differences in the soil with a piezoelectric sensor embedded in a water-filled porous ceramic cup. At EMM\_1, two sensors of each type (2 × VWC and 2 × SWP) were installed at depths of 0.15 m, 0.30 m, 0.50 m, and 1.00 m. At EMM\_2, two sensors of each type were installed at 0.15 m, 0.50 m, and 0.95 m, with an additional sensor pair (1 × VWC and 1 × SWP) installed at 0.20 m and 0.70 m. No site-specific calibration of the sensors was conducted, as the original study by Wicki et al., (2024) focused primarily on relative changes in VWC. While this study used absolute values to estimate effective stress, only relative changes in effective stress were analyzed for comparison with the strain rates derived from the DAS measurements.

The depths were selected to match the sensitivity of each DAS product: 0.15 m soil moisture corresponds to the fiber burial depth and was used for quasi-static strain (L330), while dv/v is sensitive to the whole soil profile and was therefore compared with a two-layer model (0–0.15 m and 0.15–1.53 m), with the deeper layer represented by 0.95 m data (L264). To avoid confusion, we added a clarifying sentence after L330.

unn

We focus on 0.15 m depth because it matches the cable burial depth and thus best represents quasi-static strain, which is different from the dv/v analysis. This is because dv/v is sensitive to seismic velocity changes integrated over

several meters depth, whereas the low-frequency strain reflects direct nearsurface deformation at the fiber depth.

unn

The spatial correlation between DAS and soil moisture metrics is shown in Fig.10f with discussion starting L440. During the monitoring period, the sensors at the depths used remained operational.

**3) dv/v temporal resolution and depth sensitivity**

dv/v is computed from daily noise stacks; such products cannot substantiate intraday variability. Also, depth sensitivity of the 8–16 Hz band should be explicitly related to the moisture sensors used.

**Requested actions:**

- State explicitly that dv/v is daily and avoid intraday claims unless sub-daily processing is added.
- Provide a concise depth-sensitivity summary for the analyzed band and reconcile with the chosen moisture depths.

The previously revised version has rephrased the sub-daily statement at L453 to: 'The long-term soil consolidation is further supported...'. The summary of depth sensitivity and the rationale for the chosen moisture depth are provided in the paragraph starting at L370.

**4) Instrument/cable low-frequency response and temperature/drift**

Thermal strain is estimated to be small and no correction is applied. Please also describe potential interrogator/cable LF response and drift controls when integrating strain-rate.

**Requested actions:**

- Specify cable construction (tight-buffer vs loose-tube, jacket, burial details) and discuss implications for LF response.
- Include a brief check comparing strain-rate with the temporal derivative of temperature over selected windows to assess instrument-related contributions.
- Summarize detrending/high-pass choices used to mitigate integration drift and how robustness was verified.

Please find an entire new section added in the manuscirpt 6.1.1 Instrumental drift quantification at around L280. In short summarize, we caluculated the instrumental drift from the aprt of the cable in the garage and also applied

thermal correction. The cable constructiona and discussion is presented at L410.

**Minor / Editorial**

• Appendix: correct "Index of fraction" → "Index of refraction."

We appreciate the comment. This issue was already modified in the earlier version.

• After Figure 3 caption: ensure the next paragraph begins with a capitalized "Hourly ...".

Thanks for your suggestion. This concerns a figure placement issue. It fits within the paragraph and breaks across a single sentence. In the last version, the starting word is 'heterogeneity,' which belongs to the last sentence and is therefore not capitalized.

- Consider adding cable specs (make/type) in Methods §3.2 for completeness. Thanks. This was added as well in the last version "fiber-optic gel-filled non-metallic loose..."
- Ensure all instances of resolution claims are consistent with the stated GL and spacing.

Thank you. We have ensured that all sub-meter phrasing has been removed and that the text now corresponds to the gauge length.

Minor language/clarity – around L180 (soil-moisture paragraph after Fig. 3) The current wording is understandable but could be tighter and more precise (e.g., SWP "decreases" → "becomes more negative"), and the historical citation after "Pearson" isn't needed here.

Thank you very much for your suggestion. We have revised the paragraph accordingly

Consider rephrasing the following sections for clarity and consistency:

• A) Results — Soil-moisture paragraph (after Fig. 3, ~L175–L186). Suggested replacement:

Soil moisture closely tracked rainfall: volumetric water content (VWC) rose during infiltration, while soil-water potential (SWP) became more negative. Measurements from EMM\_1 and EMM\_2 were highly correlated for both

variables (Pearson's r > 0.9), indicating that either site can serve as a representative indicator of regional soil-moisture dynamics.

Thank you very much for your suggestion. We have revised the paragraphs accordingly.

• B) Methods — Soil moisture (Section 4.2 "In situ Soil Moisture Measurements", ~L170–L176 starter lines; drop-in fits immediately after). Suggested replacement:

Volumetric water content (VWC) and soil-water potential (SWP) were measured at 0.15–1.0 m depth (10-min cadence) near the buried cable. We screened outliers, computed daily medians to match dv/v stacks, and used both a shallow composite (0.15–0.40 m) and a multi-depth composite (0.15–1.0 m) to assess sensitivity to depth selection.

Thank you. Please see above the detail depths, model type and correction on the soil moisture measurements. We value your suggestion and also add this sentence at the end at L156:

m

We computed daily medians to match the dv/v analysis. For low-frequency strain analysis, we used the 0.15 m depth, while the dv/v models were built using a multi-depth composite at 0.15 m and 0.95 m. Further details are provided in the following sections.

um

• C) Conclusion — resolution claim (the "In conclusion..." sentence, ~L403–L407; the number 405 appears at line end).

Suggested replacement:

We integrate traditional seismic wave analysis with continuous monitoring of quasi-static deformation using DAS to track moisture-driven hydromechanical changes, achieving meter-scale along-cable resolution with a 10 m gauge length.

Thank you very much for your suggestion. We have revised the paragraphs accordingly.